# Evaluating the predictive significance of D-dimer in conjunction with CA724 for the postoperative outcomes in gastric cancer: A retrospective cohort analysis

**Yuanzheng Zhao[1], Jiaqi Han[1], Rong Yang[2], Shuqin Wang[1], Xinran Zhao[1], Ziyuan Wang[1], Hongxia Lu [2]\***

**1** Fenyang College Shanxi Medical University, Shanxi, China; No.16 College Road, Lvliang City, Shanxi Province, China, **2** Department of Gastroenterology, Shanxi Province Cancer Hospital/ Shanxi Hospital Affiliated to Cancer Hospital, Chinese Academy of Medical Sciences/Cancer Hospital Affiliated to Shanxi Medical University, Shanxi, China; No. 3, Zhigong New Street, Taiyuan City, Shanxi Province, China

\* luhongxia@sxmu.edu.cn

## Abstract

### Background

Gastric cancer represents a highly aggressive form of malignant tumor originating from the epithelial cells lining the gastric mucosa. Despite notable improvements in treatment approaches over the last few years, the associated mortality rate continues to be considerably high. Therefore, there is a pressing requirement for dependable biomarkers that can be utilized to predict and monitor prognosis, as well as to formulate targeted treatment strategies for patient groups at high risk.

### Methods

We conducted an analysis of data collected from patients who were diagnosed with gastric cancer and underwent radical gastrectomy at Shanxi Cancer Hospital from June 2017 to June 2018, with follow-up data gathered over a five-year duration until 2023. Patient follow-up information was sourced from the hospital's monitoring system. The analysis focused on the variances in effectiveness of D-dimer against different tumor markers through Cox stratification analysis. The tumor marker that exhibited the most pronounced impact was selected to formulate a novel combined indicator. Furthermore, we examined how this combined indicator influences five-year overall survival (OS) outcomes following gastric cancer surgery using Cox multivariate regression analysis.

### Results

The Cox multivariate regression analysis revealed that the effect value of the D_Dimer-CA724 Middle group on the overall survival rate post-surgery for gastric cancer was found to be 1.42 (1.13–1.78), p = 0.003 (<0.05), in comparison with the D_Dimer-CA724 Low group. For the D_Dimer-CA724 High group, the effect value on overall survival after gastric cancer surgery was 2.11 (1.65–2.68), p < 0.001. Additionally, the

**Data availability statement:** All relevant data are within the manuscript and its Supporting Information files.

**Funding:** The author(s) received no specific funding for this work.

**Competing interests:** The authors have declared that no competing interests exist.

trend test results indicated a value of 1.46 (1.29–1.64) with p < 0.001, demonstrating statistical significance. When compared to the D_Dimer-CA724 Low group, both the D_Dimer-CA724 Middle and High groups showed markedly poorer prognoses, with increased risks of 42% and 111%, respectively, highlighting a highly significant finding in clinical practice.

## Conclusion

The integrated measure of D-dimer and CA724, referred to as D-dimer_CA724, serves as an independent predictor for the postoperative outcomes of gastric cancer, demonstrating superior predictive capability compared to the individual markers. In clinical settings, patients with gastric cancer exhibiting elevated levels of D-dimer_CA724 tend to experience worse prognoses following surgery. This measure holds significant potential for widespread application and promotion within clinical practice.

## Introduction

Cancers of the digestive tract, such as esophageal cancer, gastric cancer, and colorectal cancer, are prevalent malignant tumors worldwide and are linked to high rates of mortality [1]. As reported in the Global Cancer Statistics for 2020, there were 19.3 million new cases of cancer and 10 million deaths linked to the disease. Within these statistics, colorectal cancer has the third-highest incidence rate at 10%, while it is the second leading cause of cancer-related deaths at 9.4%. Gastric cancer, on the other hand, ranks fifth in incidence at 5.6% and fourth in mortality rate at 7.7% [2]. The classification system used by the World Health Organization divides tumors of the digestive system into multiple categories, which include esophageal cancer, gastric cancer, small intestine cancer, hepatocellular carcinoma, gallbladder cancer, bile duct cancer, pancreatic cancer, and colorectal cancer [3]. These tumors have the highest rates of incidence and mortality among all cancer types [4], making them the foremost cause of fatalities related to cancer [5]. Gastric cancer (GC), especially, is a highly aggressive malignancy that arises from the epithelial cells of the gastric mucosa [6]. It ranks among the most aggressive tumors within the digestive tract, exhibiting a considerable capacity for invasion [7] and contributing to a high incidence and mortality rate globally, which is responsible for about 10% of all cancer-related deaths worldwide, thereby posing a significant challenge to global health [8–12]. Although there has been a reduction in incident cases and improvements in treatment, the mortality rate for gastric cancer continues to be substantial [13,14], underscoring the urgent need for targeted efforts in prognostic prediction and the discovery of biomarkers aimed at improving patient outcomes.

D-dimer, a product resulting from the breakdown of fibrin, serves as a standard biomarker for coagulation. Elevated levels of this marker are indicative of a hypercoagulable condition [15,16]. Research indicates that tumor cells secrete substances that either promote coagulation or facilitate fibrinolysis, leading to excessive coagulation. This process is thought to attract platelets and support tumor advancement [17–19]. Prior research has shown a link between increased D-dimer levels and unfavorable prognosis across various cancer types, especially in gastric cancer [20–22]. In recent studies, the research groups led by Go, S, Liu, L, Suzuki, T, Dai, H, and Kim EY have identified a significant relationship between heightened D-dimer levels and negative outcomes in cancer patients [23–26]. Given the ease and accuracy of measuring D-dimer, Dr. Guan Y's team has stressed the significance of tracking this marker as a dependable predictor for patients with cancer, including those diagnosed with gastric cancer [27].

CA724 is a glycoprotein with a high molecular weight, recognized as one of the most efficient tumor markers for the diagnosis of gastric cancer [28]. It is widely applied in both diagnosing and evaluating the prognosis of this condition [29–31]; nevertheless, its sensitivity is still not ideal [32]. In recent times, improvements in our comprehension of coagulation and cancer have given rise to the development of combined indicators, representing a new trend. Integrating serological indicators with tumor markers tends to show greater specificity and sensitivity when predicting the survival rates of gastric cancer patients, resulting in more meaningful outcomes [33]. In addition, CA724 and D-dimer are both indicators of routine in-hospital examinations for patients and can be easily obtained in clinical treatment. Therefore, in order to further obtain better prognostic indicators, we propose that the combination of preoperative D-dimer and CA724 could enhance the prediction of postoperative survival rates for patients with gastric cancer. However, there is a significant scarcity of studies exploring the role of preoperative D-dimer alongside CA724 in forecasting postoperative prognosis for this group of patients. Therefore, we carried out a large-scale retrospective cohort study aimed at evaluating the predictive significance of preoperative D-dimer in combination with CA724 for the long-term outcomes of gastric cancer following surgical treatment.

## Methods

### Data acquisition

Numerous investigations concerning prognosis often focus on overall survival (OS) as the main endpoint [34–36]. In the present research, the central evaluative metric is the combined D-dimer_CA724 indicator, with the ultimate outcome being the 5-year overall survival rate (OS). The measuring instrument name of CA724 selected by the Medical Laboratory Department of our hospital is "Transparent Automatic Flow Fluorescence Immunoassay Analyzer", model "TESMI F4000", and the D-dimer measuring instrument is "Automatic Coagulation Analyzer", model "CS5100". In terms of medical laboratory testing, our hospital's laboratory department regularly checks the external environment such as temperature, humidity, and air quality to ensure the quality and reliability of the testing; During sampling, strictly follow the operating procedures, reserve samples according to requirements, strictly follow the order of use of sampling tubes, and send them for inspection in a timely manner; In terms of instruments and equipment, we use the most advanced measuring instruments with high precision, sensitivity, accuracy, and stability, and complete them through automation and intelligence. We also ensure that the inspection instruments are calibrated regularly before or during use. Ensure the reliability of our specimen testing results.The objective of this study is to investigate how D-dimer_CA724 can predict long-term outcomes for gastric cancer patients post-surgery. A comprehensive collection of clinicopathological and demographic information was retrospectively gathered from the institutional database, which included details such as patient age, gender, tumor location, histological grading, and clinical TNM stage, following the TNM classification system outlined by the American Joint Committee on Cancer (AJCC 7th ed., 2010) [37]. Furthermore, data on the 5-year follow-up was extracted from the follow-up system of our hospital. Subsequently, the collected information was input into a spreadsheet (Microsoft Excel 2013, Microsoft Corporation, Redmond, WA, USA) and anonymized prior to the calculation of prognostic indices. The start and end date of this study is October 15, 2024 and November 20, 2024, which was conducted after complete ethical review approval.

### Study population

This retrospective cohort study, conducted at a single center, assessed data from individuals who had undergone gastrectomy with the intention of achieving a cure between June 2017

and June 2018 at the Shanxi Provincial Cancer Hospital, part of the Chinese Academy of Medical Sciences, located in Taiyuan, China. The main inclusion criteria for participation in this research were as follows: (1) individuals over the age of 18; (2) a confirmed diagnosis of gastric adenocarcinoma via pathological evaluation; and (3) patients who received radical gastrectomy for gastric cancer. Exclusion criteria comprised the following: (1) individuals with a prior history of other malignant tumors or multiple primary cancers; (2) individuals with specific conditions that could impair the coagulation system; (3) patients lost to follow-up early on, leading to incomplete baseline data; and (4) those diagnosed with gastric cancer types other than adenocarcinoma. In total, 1,249 patients were included in the analysis (refer to Fig 1). Coagulation tests were performed upon patient admission to the hospital. The disease explored in this study is "gastric adenocarcinoma". The pathological manifestations and biochemical indicators of other types of tumors or gastric cancer may differ, which further affects the stability of the experimental results. Therefore, this study excluded all tumor diseases except for "gastric adenocarcinoma". This study collected data randomly and generated a usable dataset through computer anonymous encoding to minimize selection bias to

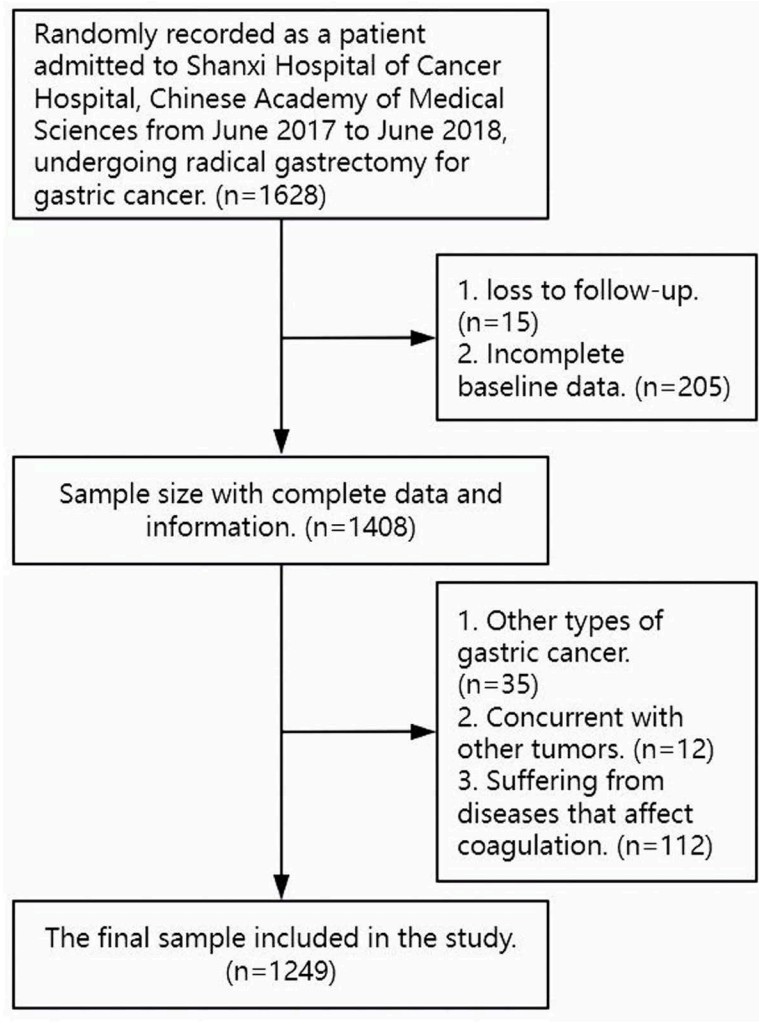

**Fig 1. Research population admission and exclusion process diagram.**

the greatest extent possible. While categorized as non-interventional clinical research due to its retrospective design, ethical approval was secured from the relevant ethics committee. The study received approval from the Shanxi Cancer Hospital ethics committee (No: KY-2024149), and the requirement for additional formal consent beyond what patients had provided before hospitalization was waived. This research complied with the principles established in the Helsinki Declaration of 1975, as amended in 2008.

### Preoperative D-dimer, CEA, CA242, CA724 and CA199

All hematological assessments were performed at our facility following established operating protocols, utilizing uniform testing equipment, standard reference ranges, and ensuring high data consistency. The laboratory tests routinely conducted included D-dimer, CEA, CA242, CA724, and CA199.

In this study, the focus is on the preoperative (initial admission) D-dimer assessment. We hypothesize that the effects of various surgical procedures and additional diagnostic tests or treatments upon admission may influence D-dimer levels to different extents, thereby impacting the accuracy of the experimental findings.

The aim was to properly group D-dimer, CEA, CA242, CA724 and CA199, so the study used the ROC curves to determine the cutoff points of them. The cutoff points were computed with the maximal Youden index. Determine the optimal cut-off points for the D-dimer, CEA, CA242, CA724 and CA199 were 122.5, 2.18, 10.14, 2.56 and 25.66, respectively based on the sensitivity, specificity, and maximum Jordan index of each indicator (Fig 2, Table 1). Subsequently, the D-dimer, CEA, CA242, CA724 and CA199 were stratified into D-dimer < 122.5 or D-dimer ≥ 122.5, CEA < 2.18 or CEA ≥ 2.18, CA242 < 10.14 or CA242 ≥ 10.14,

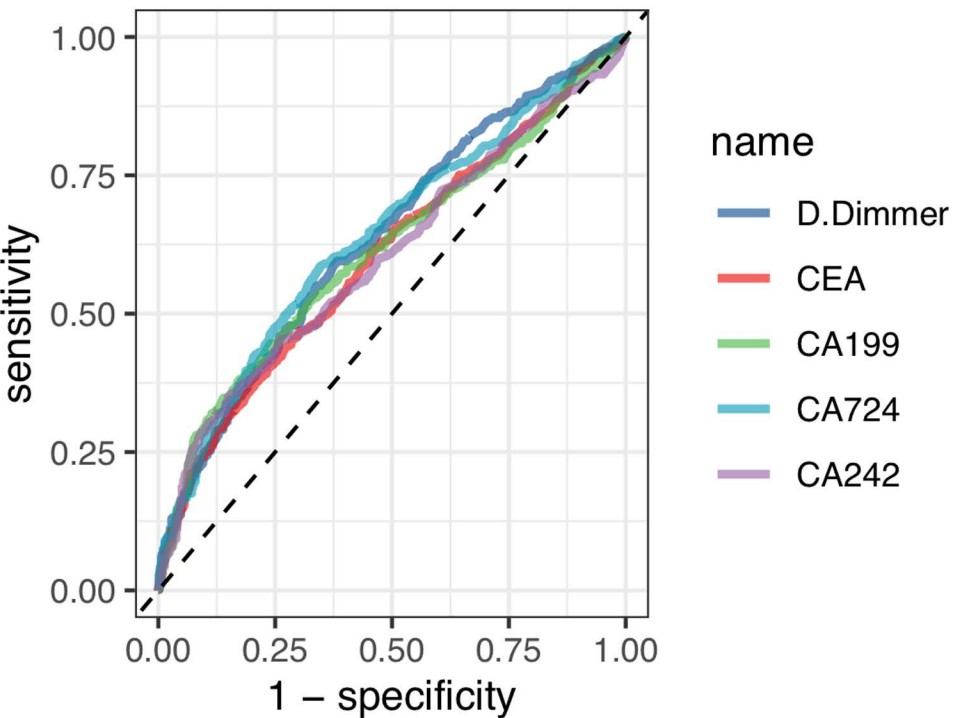

**Fig 2. ROC curves of D-dimer, CA724, CA199, CA242, CEA.**

**Table 1. The area under the curve and optimal cutoff value of the ROC curves for D-dimer, CA724, CA199, CA242, and CEA.**

| Variable | AUC(95%CI) | Specificity | Sensitivity | Truncated value |
|---|---|---|---|---|
| D.Dimmer | 63.6527% (60.5861% ~ 66.7194%) | 0.6175 | 0.5951 | 122.5 |
| CEA | 60.4802% (57.3401% ~ 63.6203%) | 0.7302 | 0.4377 | 2.18 |
| CA199 | 61.4188% (58.2727% ~ 64.5649%) | 0.8603 | 0.3475 | 25.66 |
| CA724 | 63.9606% (60.8863% ~ 67.0348%) | 0.654 | 0.5852 | 2.56 |
| CA242 | 60.3061% (57.1477% ~ 63.4646%) | 0.846 | 0.3541 | 10.14 |

CA727 < 2.56 or CA727 ≥ 2.56, CA199 < 25.66 or CA199 ≥ 25.66 for all subsequent analyses. According to the D-dimer_CA724 score, gastric cancers were divided into 3 groups: D-dimer-CA242 = 0 (D-dimer < 122.5 and CA724 < 2.56 ng/ml), D-dimer_CA724 = 1. (D-dimer < 122.5 and CA724 ≥ 2.56 ng/ml, D-dimer ≥ 122.5 and CA724 < 2.56 ng/ml) and D-dimer_CA724= 2 (D-dimer ≥ 122.5 and CA724 ≥ 2.56 ng/ml)..

## Statistical analysis

This study grouped patients based on the ultimate survival outcome of the patient. Other categorical variables were presented in terms of frequency or percentage. The lack of intersection in the K-M survival curves indicates that this is an proportional hazards model. Therefore, this study used proportional Cox regression analysis. Due to the small number of missing values and in order to avoid generating virtual sample sizes, this study chose the simple deletion method to handle missing values. Statistical tests such as χ2 test for categorical variables, normal distribution for continuous variables, and Mann-Whitney U test for skewed distribution were used to compare differences between the D-dimer_Low group and D-dimer_High group (Table 2). Adjust covariates (location, invasion vessel, size, AJCC grading) based on their statistical performance and clinical experience. Using Cox stratification analysis to explore the difference in the impact of D-dimer levels on overall survival (OS) after gastric cancer surgery in various subgroups including CEA, CA242, CA724, and CA199, and drawing a forest plot (Table 3, Fig 3). The difference in efficacy between the D-dimer groups was **20%** in the **CA724** subgroup; 6% between CA199 subgroups; 8% between CA242 subgroups (There was no statistical significance in the CEA subgroups). Select one groups with the greatest difference in the D-dimer groups (D-dimer and CA724) as a new indicator. Exploring the effect of D-dimer and the combined index of CA724 on overall survival (OS) after gastric cancer surgery using Cox multivariate regression analysis (Table 4).

All the analyses were performed with the statistical software packages R 3.3.2 (http://www.R-project.org, The R Foundation) and Free Statistics software versions 1.1.8 A two-tailed test was performed and p < 0.05 was considered statistically significant. P values less than 0.05 (two-sided) were considered statistically significant. The continuous variables are presented as the mean ± standard deviation (SD), while nominal variables are presented as the total number and percentage. If the HR value of the effect value is more than 1 and the 95% confidence interval does not cross 1, it indicates predicting poor postoperative prognosis for gastric cancer.

Finally, plot the K-M survival curves for D-dimer, CA724, and D-dimer-CA724, respectively, to visually demonstrate the impact of individual and combined indicators on postoperative prognosis of gastric cancer. (Fig 4, Fig 5, Fig 6)

**Table 2. Baseline data table of study population presented in groups of D-dimer.**

| Variables | Total (n = 1249) | D-Dimer-Low (n = 641) | D-Dimer-High (n = 608) | p |
|---|---|---|---|---|
| **Gender, n (%)** | | | | **<0.001** |
| Female | 238 (19.1) | 96 (15) | 142 (23.4) | |
| Male | 1011 (80.9) | 545 (85) | 466 (76.6) | |
| **Old, Mean ± SD** | 59.9 ± 9.9 | 57.9 ± 9.4 | 61.9 ± 10.1 | **<0.001** |
| **Smoke, n (%)** | | | | **0.001** |
| No smoking | 678 (54.3) | 320 (49.9) | 358 (58.9) | |
| Smoking | 571 (45.7) | 321 (50.1) | 250 (41.1) | |
| **Drink, n (%)** | | | | 0.064 |
| No drinking | 1039 (83.2) | 521 (81.3) | 518 (85.2) | |
| Drinking | 210 (16.8) | 120 (18.7) | 90 (14.8) | |
| **Glycuresis, n (%)** | | | | 0.556 |
| No glycuresis | 1154 (92.4) | 595 (92.8) | 559 (91.9) | |
| Glycuresis | 95 (7.6) | 46 (7.2) | 49 (8.1) | |
| **Hypertension, n (%)** | | | | 0.188 |
| No hypertension | 987 (79.0) | 516 (80.5) | 471 (77.5) | |
| Hypertension | 262 (21.0) | 125 (19.5) | 137 (22.5) | |
| **Cardiovascular.and. cerebrovascular.diseases, n (%)** | | | | 0.955 |
| Yes | 1153 (92.3) | 592 (92.4) | 561 (92.3) | |
| No | 96 (7.7) | 49 (7.6) | 47 (7.7) | |
| **Location, n (%)** | | | | 0.516 |
| Proximal | 666 (54.4) | 354 (55.9) | 312 (52.7) | |
| Middle | 234 (19.1) | 118 (18.6) | 116 (19.6) | |
| Distal | 325 (26.5) | 161 (25.4) | 164 (27.7) | |
| **Size, Mean ± SD** | 4.8 ± 2.6 | 4.3 ± 2.4 | 5.3 ± 2.7 | **<0.001** |
| **Lauren, n (%)** | | | | 0.449 |
| Intestinal | 216 (33.6) | 111 (35.1) | 105 (32.1) | |
| Diffuse | 202 (31.4) | 102 (32.3) | 100 (30.6) | |
| Mixed | 225 (35.0) | 103 (32.6) | 122 (37.3) | |
| **Degree, n (%)** | | | | 0.188 |
| Moderately | 310 (28.1) | 173 (30.5) | 137 (25.6) | |
| Moderately to highly | 326 (29.6) | 164 (28.9) | 162 (30.3) | |
| Highly | 466 (42.3) | 230 (40.6) | 236 (44.1) | |
| **AJCC.state, n (%)** | | | | **<0.001** |
| I | 219 (18.9) | 137 (23) | 82 (14.6) | |
| II | 269 (23.2) | 162 (27.2) | 107 (19) | |
| III | 622 (53.7) | 275 (46.1) | 347 (61.7) | |
| IV | 48 (4.1) | 22 (3.7) | 26 (4.6) | |
| **Vessel.invasion, n (%)** | | | | **0.04** |
| No Invasion | 648 (55.3) | 352 (58.2) | 296 (52.2) | |
| Invasion | 524 (44.7) | 253 (41.8) | 271 (47.8) | |
| **Neural.invasion, n (%)** | | | | 0.717 |
| No Invasion | 707 (60.3) | 368 (60.8) | 339 (59.8) | |
| Invasion | 465 (39.7) | 237 (39.2) | 228 (40.2) | |
| **Surgical.type, n (%)** | | | | **<0.001** |
| Laparoscope surgery | 475 (38.2) | 261 (40.7) | 214 (35.6) | |

*(Continued)*

**Table 2.** (Continued)

| Variables | Total (n = 1249) | D-Dimer-Low (n = 641) | D-Dimer-High (n = 608) | p |
|---|---|---|---|---|
| Traditional surgery | 735 (59.2) | 374 (58.3) | 361 (60.1) | |
| Non radical surgery | 32 (2.6) | 6 (0.9) | 26 (4.3) | |
| **Postoperative.chemotherapy, n (%)** | | | | **<0.001** |
| No chemotherapy | 682 (55.4) | 319 (50.3) | 363 (60.7) | |
| chemotherapy | 550 (44.6) | 315 (49.7) | 235 (39.3) | |
| **T.FR.modify, n (%)** | | | | **<0.001** |
| T1 | 180 (15.5) | 119 (19.9) | 61 (10.8) | |
| T2 | 103 (8.9) | 56 (9.4) | 47 (8.3) | |
| T3 | 344 (29.7) | 160 (26.8) | 184 (32.7) | |
| T4 | 533 (45.9) | 262 (43.9) | 271 (48.1) | |
| **N.FR.modify, n (%)** | | | | **<0.001** |
| N0 | 417 (35.0) | 260 (42.1) | 157 (27.4) | |
| N1 | 187 (15.7) | 97 (15.7) | 90 (15.7) | |
| N2 | 233 (19.6) | 117 (18.9) | 116 (20.2) | |
| N3 | 353 (29.6) | 143 (23.1) | 210 (36.6) | |
| **CEA, n (%)** | | | | **<0.001** |
| <2.18 | 803 (64.8) | 440 (69.2) | 363 (60.1) | |
| ≥2.18 | 437 (35.2) | 196 (30.8) | 241 (39.9) | |
| **CA199, n (%)** | | | | **<0.001** |
| <25.66 | 941 (75.9) | 510 (80.2) | 431 (71.4) | |
| ≥25.66 | 299 (24.1) | 126 (19.8) | 173 (28.6) | |
| **CA242, n (%)** | | | | **0.001** |
| <10.14 | 927 (74.8) | 503 (79.1) | 424 (70.2) | |
| ≥10.14 | 313 (25.2) | 133 (20.9) | 180 (29.8) | |
| **CA724, n (%)** | | | | **<0.001** |
| <2.56 | 667 (53.8) | 375 (59) | 292 (48.3) | |
| ≥2.56 | 573 (46.2) | 261 (41) | 312 (51.7) | |

## Results

As explained above, A total of 1249 participants were included in the final data analysis, as illustrated (Fig 1). Baseline characteristics of these participants were presented in Table 1 based on D-dimer Low group and D-dimer High group. On average, the selected participants were 59.9 ± 9.9 years old, with approximately 80.9% being male (Table 2).

Through receiver operating characteristic (ROC) curve, the optimal critical values (122.5, 2.18, 10.14, 2.56 and 25.66) for D-dimer, CEA, CA242, CA724 and CA199, and design them into each subgroup based on these critical values. After stratifying D-dimer in different subgroups using Cox stratification analysis, it was found that: the difference in impacts between the D-dimer groups was **20%** in the **CA724** subgroup; 6% between CA199 subgroups; 8% between CA242 subgroups (There was no statistical significance in the CEA subgroups). The results are visually displayed in the form of a forest figure (Table 3, Fig 3). Among them, the preoperative D-dimer showed the most significant inter group differences in the CA724 subgroup. Based on the statistical results, generate a new variable, D-dimer_CA724.

Cox multivariate regression analysis showed that, the effect value of the D_Dimer-CA724 Middle group on the overall survival rate of gastric cancer after surgery was 1.42 (1.13–1.78),

**Table 3. Cox subgroup analysis table of D-dimer in CA724, CA199, CA242, CEA subgroups.**

| Subgroup | Variable | n.total | n.event_% | Followup.Time | adj.HR_95 CI | adj.P_value |
|---|---|---|---|---|---|---|
| **CA724-Low** | | | | | | |
| | D.Dimer-Low | 375 | 118 (31.5) | 18152 | 1(Ref) | |
| | D.Dimer-High | 292 | 136 (46.6) | 11873 | **1.42 (1.09 ~ 1.85)** | 0.009 |
| **CA724-High** | | | | | | |
| | D.Dimer-Low | 261 | 129 (49.4) | 11020 | 1(Ref) | |
| | D.Dimer-High | 312 | 227 (72.8) | 8848 | **1.62 (1.28 ~ 2.04)** | |
| **CEA-Low** | | | | | | |
| | D.Dimer-Low | 440 | 161 (36.6) | 20568 | 1(Ref) | |
| | D.Dimer-High | 363 | 182 (50.1) | 14234 | 1.19 (0.95 ~ 1.5) | 0.13 |
| **CEA-High** | | | | | | |
| | D.Dimer-Low | 196 | 86 (43.9) | 8604 | 1(Ref) | |
| | D.Dimer-High | 241 | 181 (75.1) | 6487 | 2.2 (1.67 ~ 2.91) | |
| **CA199-Low** | | | | | | |
| | D.Dimer-Low | 510 | 169 (33.1) | 24649 | 1(Ref) | |
| | D.Dimer-High | 431 | 230 (53.4) | 16375 | 1.54 (1.25 ~ 1.9) | |
| **CA199-High** | | | | | | |
| | D.Dimer-Low | 126 | 78 (61.9) | 4523 | 1(Ref) | |
| | D.Dimer-High | 173 | 133 (76.9) | 4346 | 1.48 (1.09 ~ 2.02) | 0.012 |
| **CA242-Low** | | | | | | |
| | D.Dimer-Low | 503 | 168 (33.4) | 24184 | 1(Ref) | |
| | D.Dimer-High | 424 | 226 (53.3) | 15877 | 1.55 (1.25 ~ 1.92) | |
| **CA242-High** | | | | | | |
| | D.Dimer-Low | 133 | 79 (59.4) | 4988 | 1(Ref) | |
| | D.Dimer-High | 180 | 137 (76.1) | 4844 | 1.47 (1.09 ~ 1.98) | 0.012 |

$p = 0.003$ (<0.05), compared to the D_Dimer-CA724 Low group; the effect value of the D_Dimer-CA724 High group on the overall survival rate after gastric cancer surgery was 2.11 (1.65–2.68), $p < 0.001$; the results of the trend test were 1.46 (1.29–1.64) $p < 0.001$, indicating statistical significance. Compared with the D_Dimer-CA724 Low group, the D_Dimer-CA724 Middle group and the D_Dimer-CA724 High group have significantly worse prognosis, with an increased risk of **42%** and **111%**, respectively, which is a very significant result in clinical practice. And the effect value of the trend test is 1.46, indicating that as the group increases, the postoperative prognosis of gastric cancer shows a decreasing trend (Table 4).

The K-M survival curves were plotted by grouping D-dimer, CA724, and D-dimer-CA724 separately. The results showed that the survival rate of the high D-dimer group was significantly lower than that of the low D-dimer group; the survival rate of the CA724 high group was significantly lower than that of the low group; the survival rate of the D-dimer_CA724 groups was more significant than that of separate D-dimer and CA724 groups. (Fig 4, Fig 5, Fig 6)

## Discussion

Our research has shown that the combination of D-dimer and CA724 (D-dimer-CA724) is a significant predictor of postoperative outcomes in gastric cancer, demonstrating effects that range from 42% to 111%, which means that compared with the D-dimer CA724 Low group, the D-dimer CA724 Middle group and the D-dimer CA724 High group have protective effects on 5-year overall survival rates of 42% and 111%, respectively with a consistent upward trend

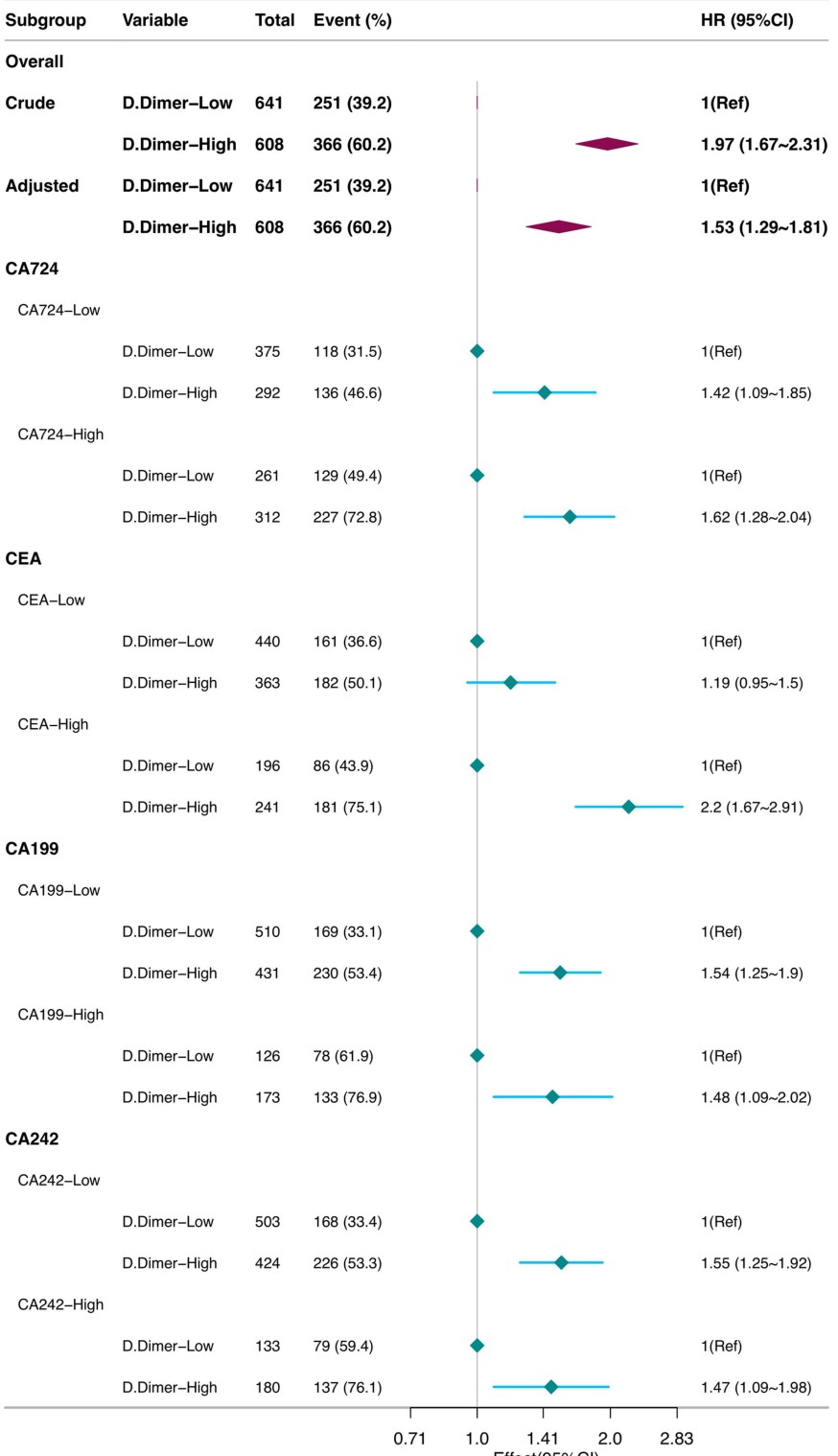

**Fig 3. Forest plot of Cox subgroup analysis.**

**Table 4. Cox multivariate regression analysis table.**

| Variable | n.total | n.event_% | Followup.Time | adj.HR_95 CI | adj.P_value |
|---|---|---|---|---|---|
| D_Dimer-CA724_Low | 375 | 118 (31.5) | 18152 | 1(Ref) | |
| D_Dimer-CA724_Middle | 553 | 265 (47.9) | 22893 | 1.42 (1.13 ~ 1.78) | 0.003 |
| D_Dimer-CA724_High | 312 | 227 (72.8) | 8848 | 2.11 (1.65 ~ 2.68) | <0.001 |
| Trend.test | 1240 | 610 (49.2) | 50161 | 1.46 (1.29 ~ 1.64) | <0.001 |

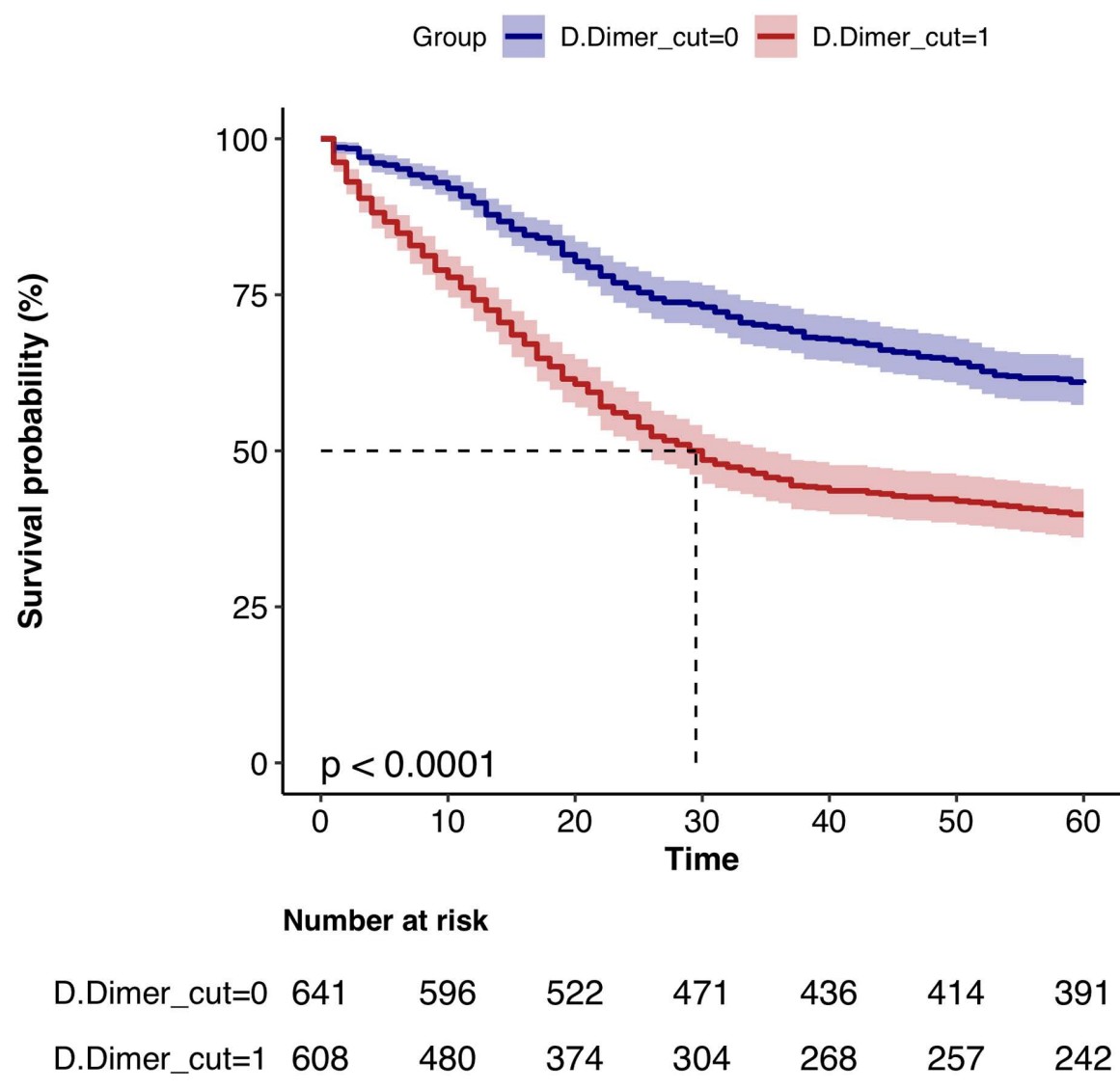

**Fig 4. K-M survival curves of D-dimer high and low groups.**

observed across different groups [1.46 (1.29–1.64), p < 0.001]. D-dimer, an established end product of fibrin breakdown, plays a role in promoting tumor angiogenesis, invasion, metastasis, and growth by influencing the migration of endothelial and cancer cells [38]. Moreover, the progression of tumors is closely linked to coagulation, as tumor growth and invasiveness

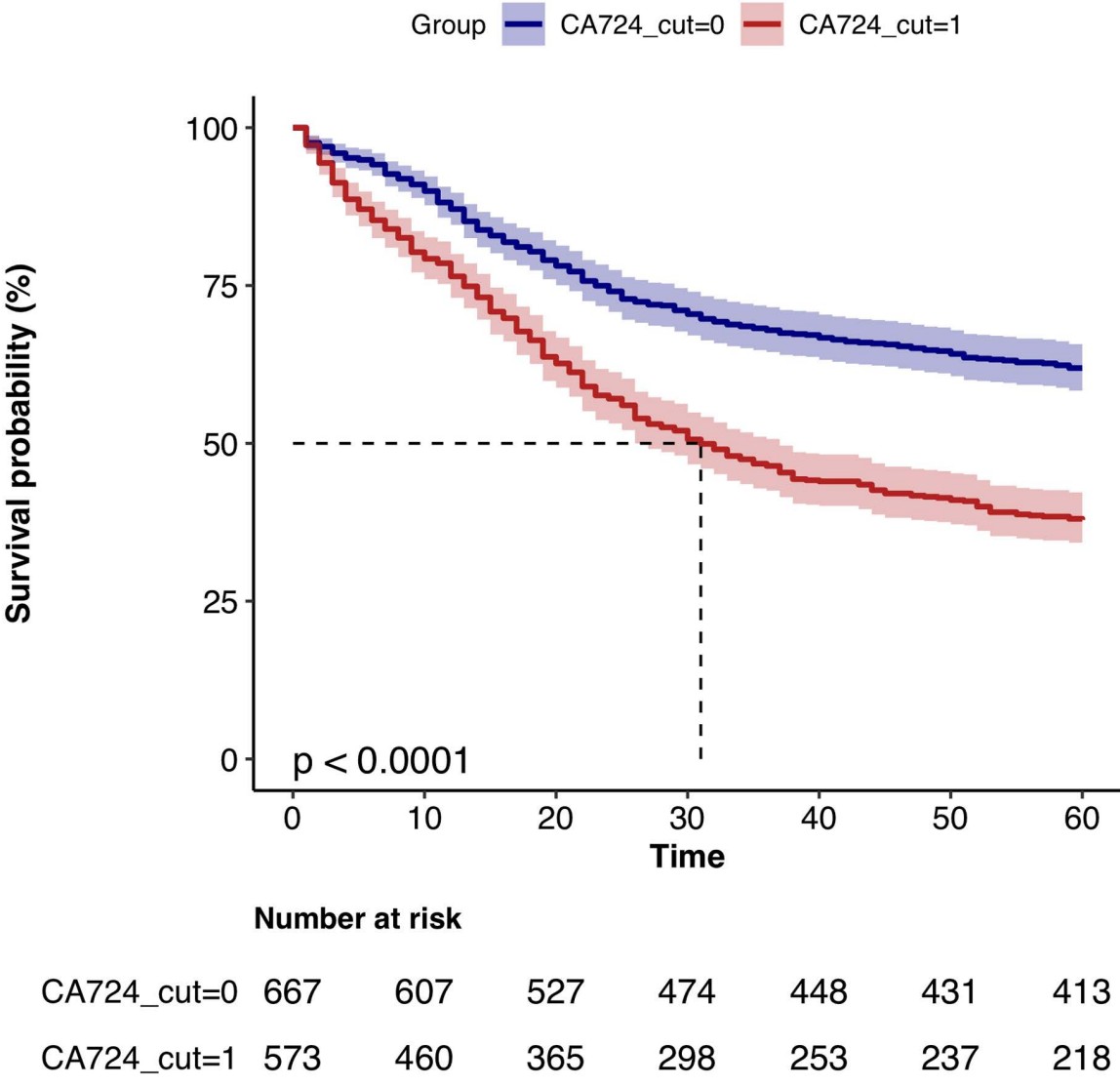

**Fig 5. K-M survival curves of CA724 high and CA724 low groups.**

depend largely on the capacity of cancer cells to facilitate angiogenesis and metastasis [19]. As a result, the initial blockage of the vascular system in distant organs by cancer cells, coagulation plasma, and platelets occurs through thrombus formation, which aids in the adhesion and dissemination of cancer cells to the walls of blood vessels [39]. It is believed that D-dimer may enhance tumor growth and invasion through interleukin-6 and vascular endothelial growth factor signaling pathways in cancer patients [40]. Additionally, earlier studies suggest that anticoagulant therapy might help extend the survival of patients with cancer [41]. Hence, D-dimer is regarded as a prognostic biomarker for gastric cancer, corroborating our research findings. CA724 has traditionally been viewed as one of the most sensitive tumor markers for gastric cancer, playing an essential role in both diagnosis and prognostic assessments, and is extensively used in clinical settings [28–31]. Nevertheless, its sensitivity is not entirely adequate [32]. In summary, D-dimer indicates more about the blood environment status of gastric cancer patients, while CA724 indicates more about the tumor microenvironment status

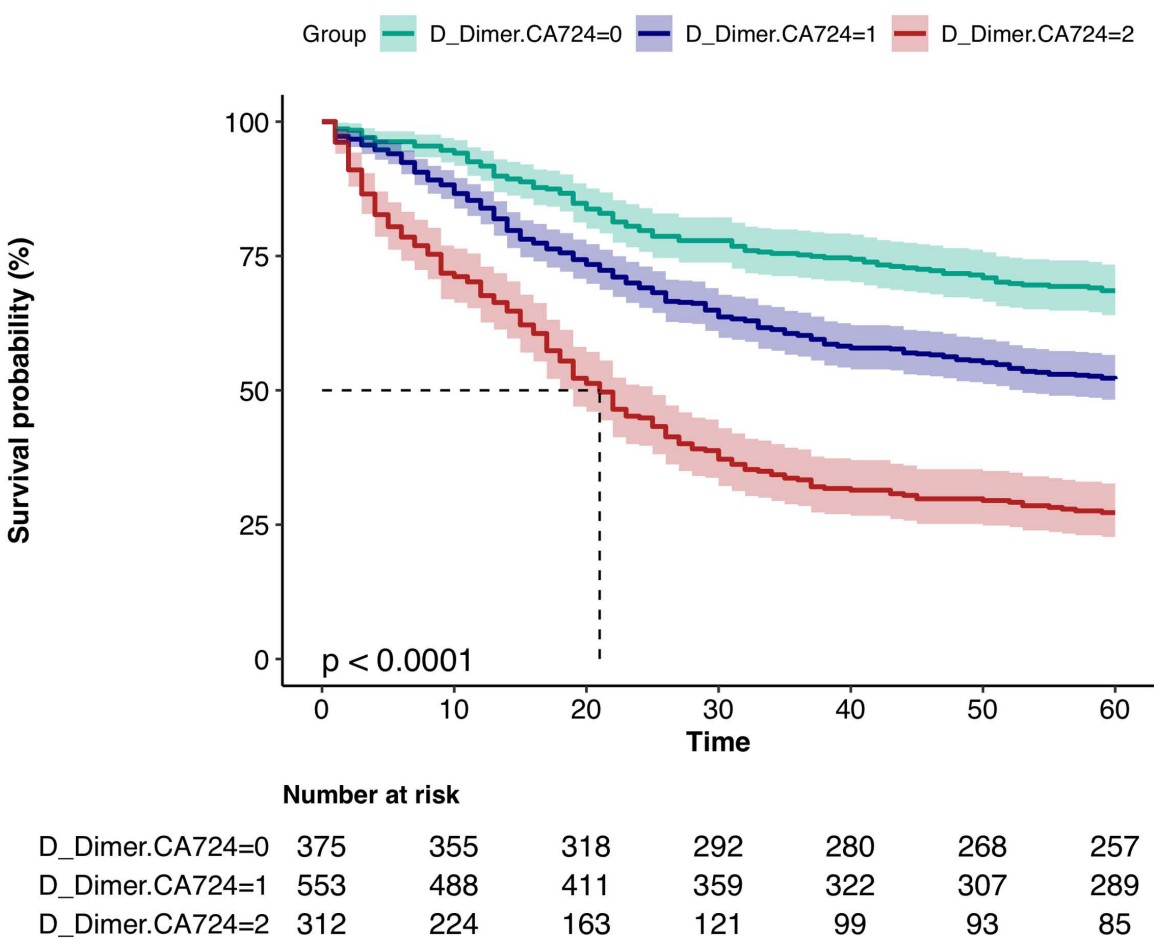

**Fig 6. K-M survival curves of D-dimer_CA724 groups.**

of patients. Therefore, combining the two will better indicate the pathological environment in which the tumor is located.Therefore, we proposed that the joint use of D-dimer and CA724 would provide improved predictive capabilities for gastric cancer compared to each marker individually, a hypothesis that our experimental findings have validated.

Eun Young Kim and colleagues examined the relationship between preoperative D-dimer levels and overall survival (OS) after gastric cancer surgery through multiple regression analysis in a cohort of 666 patients from Korea. The findings indicated an effect value of 1.70 (0.82–3.55) with a p-value of 0.153, suggesting that the significance of this effect was not substantial [26]. Similarly, Xin Zhang's team demonstrated in a retrospective propensity score matching analysis that elevated D-dimer levels serve as independent risk factors for unfavorable outcomes in gastric cancer [42]. In contrast, the joint indicator of D-dimer and CA724 examined in this research exhibited a stronger correlation and greater statistical significance, with values of 2.11 (1.65–2.68) and $p < 0.001$, compared to earlier studies. Previous studies have shown that both D-dimer and CA724 indicate poor prognosis for gastric cancer, which is similar to our research findings, indicating poor prognosis for gastric cancer. However, our research results also indicate that the combination of D-dimer and CA724 has higher effect values and smaller p-values, which is more significant and statistically significant compared to D-dimer and CA724 alone.

In conclusion, we propose that the combined preoperative measurements of D-dimer and CA724 provide enhanced predictive capability for postoperative prognosis in gastric cancer, relative to each marker evaluated alone. These biomarkers are commonly employed in clinical settings to predict postoperative outcomes and aid in the diagnosis of gastric cancer. Generally, patients with gastric cancer who show increased levels of D-dimer and CA724 are likely to face a worse prognosis after surgical intervention. Therefore, we suggest that clinical doctors should pay attention to the initial levels of D-dimer and CA724 in each hospitalized gastric cancer patient and conduct a comprehensive evaluation. Patients with high levels of D-dimer and CA724 usually indicate worse tumor blood and microenvironment status, and require more attention during treatment and follow-up. This insight may inform innovative strategies for postoperative adjuvant therapy in gastric cancer patients, including the potential use of anticoagulant treatment, aimed at optimizing survival and promoting advancements in precision oncology.

## Conclusion

The integrated measure of D-dimer and CA724, referred to as D-dimer_CA724, serves as a standalone risk factor for the postoperative outcomes in gastric cancer, demonstrating superior predictive ability compared to each individual marker. In a clinical context, patients with gastric cancer exhibiting elevated levels of D-dimer_CA724 typically encounter a less favorable prognosis following surgical intervention. This metric holds significant potential for advancement and application within clinical settings.

## Limitations of this study

The gastric cancers selected for this study were all classified as gastric adenocarcinomas; other tumor types, such as gastric neuroendocrine carcinoma and gastric squamous cell carcinoma, were not included, which may result in the unsuitability of this study to other types of tumors. Therefore, future clinical studies specifically targeting other types of tumors are needed to explore the prognosis of this particular tumor type. 2. This study only investigated that gastric cancer patients with high D-Dimer_CA724 usually face poorer prognosis after surgery, providing a new direction for future adjuvant therapy for gastric cancer. However, future prospective research is needed to explore targeted postoperative adjuvant therapy methods for this group of patients. 3. Although the sample size of this study is large enough, it is still a single center study and may have regional limitations. In the future, larger scale multi center studies will be needed for validation.

## Supporting information

**S1 Data. Research data.**
(XLSX)

## Acknowledgements

We thank Jie Liu, PhD (Department of Vascular and Endovascular Surgery, Chinese PLA General Hospital) for his helpful review and comments regarding the manuscript.

## Author contributions

**Data curation:** Yuanzheng Zhao, Rong Yang, Shuqin Wang, Xinran Zhao, Ziyuan Wang.

**Formal analysis:** Yuanzheng Zhao.

**Investigation:** Yuanzheng Zhao.

**Methodology:** Yuanzheng Zhao.

**Project administration:** Hongxia Lu.

**Resources:** Hongxia Lu.

**Software:** Yuanzheng Zhao, Jiaqi Han.

**Visualization:** Yuanzheng Zhao.

**Writing – original draft:** Yuanzheng Zhao.

**Writing – review & editing:** Hongxia Lu.

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
