## [Decision Letter · Decision Letter 0]

6 Jan 2025

PONE-D-24-54059Evaluating the predictive significance of D-dimer in conjunction with CA724 for the postoperative outcomes in gastric cancer: A retrospective cohort analysis.PLOS ONE

Dear Dr. lu,

Thank you for submitting your manuscript to PLOS ONE. After careful consideration, we feel that it has merit but does not fully meet PLOS ONE’s publication criteria as it currently stands. Therefore, we invite you to submit a revised version of the manuscript that addresses the points raised during the review process.

We look forward to receiving your revised manuscript.

Kind regards,

Xingliang Dai

Academic Editor

PLOS ONE

Journal Requirements:

2. Please include captions for your Supporting Information files at the end of your manuscript, and update any in-text citations to match accordingly. Please see our Supporting Information guidelines for more information: http://journals.plos.org/plosone/s/supporting-information .

Additional Editor Comments:

Please provide experimental validation, such as immunohistochemistry of clinical specimens.

Reviewers' comments:

Reviewer's Responses to Questions

**Comments to the Author**

1. Is the manuscript technically sound, and do the data support the conclusions?

Reviewer #1: Yes

Reviewer #2: Yes

2. Has the statistical analysis been performed appropriately and rigorously? 

Reviewer #1: Yes

Reviewer #2: Yes

3. Have the authors made all data underlying the findings in their manuscript fully available?

Reviewer #1: Yes

Reviewer #2: Yes

4. Is the manuscript presented in an intelligible fashion and written in standard English?

Reviewer #1: Yes

Reviewer #2: No

5. Review Comments to the Author

Reviewer #1: 1. **Research Background and Importance:**

- The article should provide a more detailed description of the global and regional epidemiology of gastric cancer, as well as the importance of D-dimer and CA724 as biomarkers in the prognosis of gastric cancer, to emphasize the relevance of the study.

2. **Patient Selection and Exclusion Criteria:**

- The authors need to detail the specific criteria for patient selection and exclusion, including why certain patients (e.g., those with a history of other malignant tumors) were excluded, and how these criteria affect the generalizability of the study results.

3. **Accuracy and Reliability of Biomarker Measurement:**

- The article should provide detailed information on the methods of measuring D-dimer and CA724, including the reagents and instruments used, as well as quality control measures during the measurement process, to ensure the accuracy and reliability of the results.

4. **Rationality of Statistical Analysis Methods:**

- The authors are requested to describe in detail the statistical methods used, including the selection of models, handling of variables, and how missing data are addressed. Additionally, the hypothesis testing for the Cox regression model should also be explained in detail.

5. **Image Clarity and Quality:**

- Ensure that all provided images (such as ROC curves, forest plots, survival curves, etc.) have sufficient resolution and clarity to display details clearly in both print and electronic formats.

6. **Depth of Discussion Section:**

- The authors mention that the D-dimer_CA724 combined indicator has predictive value in the prognosis of gastric cancer. Please compare this finding in detail with previous research results, especially those that explore D-dimer or CA724 alone as prognostic indicators. For example, if previous studies have shown that elevated D-dimer levels are associated with poor gastric cancer prognosis, do the findings of this study align with these results? If not, the authors need to explain the possible reasons.

Reviewer #2: This manuscript primarily investigates the prognostic value of the D-dimer and CA724 combined index in predicting postoperative outcomes in gastric cancer patients. Through a retrospective analysis of data from 1249 gastric cancer patients who underwent radical gastrectomy, the study found that the D-dimer_CA724 combined index could more accurately predict the five-year survival rate compared to individual biomarkers. The results showed that patients in the high D-dimer_CA724 group had significantly poorer postoperative outcomes, with a markedly increased risk of death, suggesting that this combined index has broad potential for clinical application. The manuscript demonstrates innovation and clinical value in the research of gastric cancer prognostic biomarkers. However, there are still some shortcomings in aspects such as spelling and grammar, scientific rigor, discussion, and conclusion. It is hoped that the authors will address the following points to improve the likelihood of the paper's acceptance.

1. The grouping method (based on the cutoff values for D-dimer and CA724) should be explained in more detail, including how these cutoff values were determined through ROC curves, and provide specific AUC values along with sensitivity and specificity data.

2. It should be clearly stated whether blinding was used during data collection to avoid subjective bias.

3. The results section should more thoroughly discuss the statistical significance of differences between groups and explain the clinical implications of these differences.

4. The impact difference between the D-dimer groups in the CA724 subgroup was 20%, between the CA199 subgroups was 6%, and between the CA242 subgroups was 8%. While CA724, CA199, and CA242 are all significant, it may be insufficient to consider only CA724 for subgroup analysis. Could a combined analysis of all three markers be conducted?

5. The discussion should delve deeper into the mechanisms underlying the combined use of D-dimer and CA724, rather than merely citing previous studies. For example, the authors could explore how these two markers might synergistically affect the prognosis of gastric cancer.

6. The clinical implications of the research findings should be more clearly stated, such as suggesting a prognostic assessment process or treatment recommendations based on the D-dimer_CA724 index.

7. Although the paper mentions the limitations of the study, these should be discussed in more detail, particularly how they might influence the study’s results, and potential solutions or future improvements in research should be suggested. Since this is a single-center study, it would also be useful to explore whether the findings are applicable in multi-center settings.

8. Check the paper language and make sure all language errors have been fixed.

6. PLOS authors have the option to publish the peer review history of their article (what does this mean? ). If published, this will include your full peer review and any attached files.

**Do you want your identity to be public for this peer review?** For information about this choice, including consent withdrawal, please see our Privacy Policy .

Reviewer #1: No

Reviewer #2: No

---

## [Author Response · Author response to Decision Letter 1]

10 Jan 2025

Reviewer1.

Thank you for your contribution to our manuscript! Your review suggestions have greatly helped improve the professionalism and detail of the manuscript! Based on your review of our manuscript, we have made the following corrections:

1.Thank you for your valuable suggestion! We have added the following epidemiological description of gastric cancer and supplements in the introduction:

As reported in the Global Cancer Statistics for 2020, there were 19.3 million new cases of cancer and 10 million deaths linked to the disease. Within these statistics, colorectal cancer has the third^^highest incidence rate at 10%, while it is the second leading cause of cancer^^related deaths at 9.4%. Gastric

cancer, on the other hand, ranks fifth in incidence at 5.6% and fourth in mortality rate at 7.7% (42).

(42) Sung H, Ferlay J, Siegel RL, Laversanne M, Soerjomataram I, Jemal estimates of incidence and mortality worldwide for 36 cancers in A and Bray F: Global cancer statistics 2020: GLOBOCAN 185 countries. CA Cancer J Clin 71: 209^^249, 2021.

In addition, CA724 and D-dimer are both indicators of routine in-hospital examinations for patients and can be easily obtained in clinical treatment. Therefore, in order to further obtain better prognostic indicators, we propose that the combination of preoperative D-dimer and CA724 could enhance the prediction of postoperative survival rates for patients with gastric cancer.

2.Thank you for your valuable suggestion! We have added the following content to 'Study Population':

The disease explored in this study is "gastric adenocarcinoma". The pathological manifestations and biochemical indicators of other types of tumors or gastric cancer may differ, which further affects the stability of the experimental results. Therefore, this study excluded all tumor diseases except for "gastric adenocarcinoma".

3.Thank you for your valuable suggestion! We have added the following content to 'Methods-Data acquisition':

The measuring instrument name of CA724 selected by the Medical Laboratory Department of our hospital is "Transparent Automatic Flow Fluorescence Immunoassay Analyzer", model "TESMI F4000", and the D-dimer measuring instrument is "Automatic Coagulation Analyzer", model "CS5100". In terms of medical laboratory testing, our hospital's laboratory department regularly checks the external environment such as temperature, humidity, and air quality to ensure the quality and reliability of the testing; During sampling, strictly follow the operating procedures, reserve samples according to requirements, strictly follow the order of use of sampling tubes, and send them for inspection in a timely manner; In terms of instruments and equipment, we use the most advanced measuring instruments with high precision, sensitivity, accuracy, and stability, and complete them through automation and intelligence. We also ensure that the inspection instruments are calibrated regularly before or during use. Ensure the reliability of our specimen testing results.

4.Thank you for your professional guidance! We have added the following content in the 'Statistical analysis' section:

Due to the small number of missing values and in order to avoid generating virtual sample sizes, this study chose the simple deletion method to handle missing values.

The lack of intersection in the K-M survival curves indicates that this is a proportional hazards model. Therefore, this study used proportional Cox regression analysis.

5.Thank you for your reminder! We will re-upload all the figures in the article.

6.Thank you for your valuable suggestion! We have added the following content in the "Discussion" section:

Previous studies have shown that both D-dimer and CA724 indicate poor prognosis for gastric cancer, which is similar to our research findings, indicating poor prognosis for gastric cancer. However, our research results also indicate that the combination of D-dimer and CA724 has higher effect values and smaller p-values, which is more significant and statistically significant compared to D-dimer and CA724 alone.

(All modifications have been highlighted prominently in the manuscript-mark )

Reviewer2.

Thank you for your contribution to our manuscript! Your review suggestions are of great help in improving the overall quality and readability of this manuscript! Based on your review of our manuscript, we have made the following corrections:

1.Thank you for your reminder! We have motified 'Table 1', in which we added the AUC, specificity, and sensitivity of D-dimer, CA724, CA199, CA242, and CEA and added the following content to the manuscript:

Determine the optimal cut-off points for the D-dimer, CEA, CA242, CA724 and CA199 were 122.5, 2.18, 10.14, 2.56 and 25.66, respectively based on the sensitivity, specificity, and maximum Jordan index of each indicator

Table 1

Variable AUC(95%CI) Specificity Sensitivity

D.Dimmer 63.6527%

(60.5861% ~ 66.7194%) 0.6175 0.5951

CEA 60.4802%

(57.3401% ~ 63.6203%) 0.7302 0.4377

CA199 61.4188%

(58.2727% ~ 64.5649%) 0.8603 0.3475

CA724 63.9606%

(60.8863% ~ 67.0348%) 0.654 0.5852

CA242 60.3061%

(57.1477% ~ 63.4646%) 0.846 0.3541

2.Thank you for your valuable suggestion! We have added the following content in the "Study Population" section:

This study collected data randomly and generated a usable dataset through computer anonymous encoding to minimize selection bias to the greatest extent possible.

3.Thank you for your professional guidance! We have added the following content in the 'Discussion' section:

“, which means that compared with the D-dimer CA724 Low group, the D-dimer CA724 Middle group and the D-dimer CA724 High group have protective effects on 5-year overall survival rates of 42% and 111%, respectively”

4.Thank you for your attention and guidance on the details of our research! Our logic here is: 'The prognostic level of gastric cancer is most significantly different and statistically significant between D-dimer and CA724 alone, so we decided to combine D-dimer and CA724 as a combined indicator.' In addition, we also validated the combination of D-dimer with CA199, CEA, CA242 in our study, and the results were either not as significant as D-dimer combined with CA724 or not statistically significant. However, if this experimental process is also included in the manuscript, it will take up a lot of space and not conform to the overall idea of the article. It may make the "results" section of the article too lengthy and affect readers' understanding of the overall logic of the manuscript. Therefore, considering the overall conciseness and ease of reader understanding of the manuscript, we ultimately chose not to display this part of the content in the final manuscript.

5.Thank you for your professional guidance! We have added the following content in the 'Discussion' section:

In summary, D-dimer indicates more about the blood environment status of gastric cancer patients, while CA724 indicates more about the tumor microenvironment status of patients. Therefore, combining the two will better indicate the pathological environment in which the tumor is located.

6.Thank you for your valuable suggestion! We have added the following content in the "Discussion" section:

Therefore, we suggest that clinical doctors should pay attention to the initial levels of D-dimer and CA724 in each hospitalized gastric cancer patient and conduct a comprehensive evaluation. Patients with high levels of D-dimer and CA724 usually indicate worse tumor blood and microenvironment status, and require more attention during treatment and follow-up.

7.Thank you for your professional guidance! We have changed section 'Limitations of this study' to '1. The gastric cancers selected for this study were all classified as gastric adenocarcinomas; other tumor types, such as gastric neuroendocrine carcinoma and gastric squamous cell carcinoma, were not included, which may result in the unsuitability of this study to other types of tumors. Therefore, future clinical studies specifically targeting other types of tumors are needed to explore the prognosis of this particular tumor type. 2. This study only investigated that gastric cancer patients with high D-Dimer_CA724 usually face poorer prognosis after surgery, providing a new direction for future adjuvant therapy for gastric cancer. However, future prospective research is needed to explore targeted postoperative adjuvant therapy methods for this group of patients. 3. Although the sample size of this study is large enough, it is still a single center study and may have regional limitations. In the future, larger scale multi center studies will be needed for validation.'

8.Thank you for reviewing the language of this manuscript. Through your review, we deeply recognize the language issues in our manuscript. We will send the manuscript for language editing and polishing after revisions

(All modifications have been highlighted prominently in the manuscript-marked)

Editor.

Thank you for your contribution to this manuscript! We uploaded our original experimental data along with the revised manuscript.

---

## [Decision Letter · Decision Letter 1]

27 Jan 2025

PONE-D-24-54059R1Evaluating the predictive significance of D-dimer in conjunction with CA724 for the postoperative outcomes in gastric cancer:

A retrospective cohort analysis.PLOS ONE

Dear Dr. lu,

Thank you for submitting your manuscript to PLOS ONE. After careful consideration, we feel that it has merit but does not fully meet PLOS ONE’s publication criteria as it currently stands. Therefore, we invite you to submit a revised version of the manuscript that addresses the points raised during the review process.

We look forward to receiving your revised manuscript.

Kind regards,

Xingliang Dai

Academic Editor

PLOS ONE

Reviewers' comments:

Reviewer's Responses to Questions

**Comments to the Author**

1. If the authors have adequately addressed your comments raised in a previous round of review and you feel that this manuscript is now acceptable for publication, you may indicate that here to bypass the “Comments to the Author” section, enter your conflict of interest statement in the “Confidential to Editor” section, and submit your "Accept" recommendation.

Reviewer #1: All comments have been addressed

Reviewer #2: All comments have been addressed

2. Is the manuscript technically sound, and do the data support the conclusions?

Reviewer #1: Yes

Reviewer #2: Yes

3. Has the statistical analysis been performed appropriately and rigorously? 

Reviewer #1: (No Response)

Reviewer #2: Yes

4. Have the authors made all data underlying the findings in their manuscript fully available?

Reviewer #1: (No Response)

Reviewer #2: Yes

5. Is the manuscript presented in an intelligible fashion and written in standard English?

Reviewer #1: (No Response)

Reviewer #2: Yes

6. Review Comments to the Author

Reviewer #1: The article, after revisions, has become richer, more rigorous, and more complete in content. The research methods and results are more credible, and the discussion section is also deeper and more insightful. The authors' serious responses to and active revisions based on the reviewers' comments have significantly enhanced the quality of the article. It is recommended to accept it for publication. However, before publication, the following points need to be noted:

1.Ensure the clarity and accuracy of all charts and graphs. The data and annotations in the charts should be consistent with the textual descriptions.

2.Conduct a final language polish to ensure smooth, accurate expression free of grammatical errors and spelling mistakes.

3.In the discussion section, further compare the results of this study with other domestic and international related research to highlight the innovations and advantages of this study. At the same time, in light of the latest research progress, provide a deeper interpretation and outlook for the research results.

Reviewer #2: The author has completely resolved my doubts, and I have no further questions. If there are no other issues, it can be accepted.

7. PLOS authors have the option to publish the peer review history of their article (what does this mean? ). If published, this will include your full peer review and any attached files.

**Do you want your identity to be public for this peer review?** For information about this choice, including consent withdrawal, please see our Privacy Policy .

Reviewer #1: No

Reviewer #2: No

---

## [Author Response · Author response to Decision Letter 2]

12 Feb 2025

Reviewer1.

Thank you for your contribution to our manuscript! Your review suggestions have greatly helped improve the professionalism and detail of the manuscript! Based on your review of our manuscript, we have made the following corrections:

1.Thank you for your reminder! We have rechecked the data in the figures, tables, and manuscript to ensure consistency, and have re uploaded them. The pixel count of the image reaches 1113 * 826, and the clarity is sufficient for readers to read clearly. You can download and see them in the attached file. (The images and tables in the main file are in preview mode and their clarity is far inferior to the original in the attachment)

2.Thank you for your reminder! We have sent the manuscript to a proofreading company for proofreading, and this time the main manuscript is the polished version. The proofreading certificate and plagiarism check certificate have also been uploaded together, please check them.

3.Thank you for your suggestion! We improved the following parts during the discussion:

“Eun Young Kim and colleagues examined the relationship between preoperative D-dimer levels and overall survival (OS) after gastric cancer surgery through multiple regression analysis in a cohort of 666 patients from Korea. The findings indicated an effect value of 1.70 (0.82-3.55) with a p-value of 0.153, suggesting that the significance of this effect was not substantial (25). Similarly, Xin Zhang's team demonstrated in a retrospective propensity score matching analysis that elevated D-dimer levels serve as independent risk factors for unfavorable outcomes in gastric cancer (41). In contrast, the joint indicator of D-dimer and CA724 examined in this research exhibited a stronger correlation and greater statistical significance, with values of 2.11 (1.65-2.68) and p<0.001, compared to earlier studies. Previous studies have shown that both D-dimer and CA724 indicate poor prognosis for gastric cancer, which is similar to our research findings, indicating poor prognosis for gastric cancer. However, our research results also indicate that the combination of D-dimer and CA724 has higher effect values and smaller p-values, which is more significant and statistically significant compared to D-dimer and CA724 alone.” Further comparison was made between relevant studies at home and abroad, and the advantages of this study were emphasized. The combined indicators had more significant effect values and statistical significance compared to individual indicators.

“In conclusion, we propose that the combined preoperative measurements of D-dimer and CA724 provide enhanced predictive capability for postoperative prognosis in gastric cancer, relative to each marker evaluated alone. These biomarkers are commonly employed in clinical settings to predict postoperative outcomes and aid in the diagnosis of gastric cancer. Generally, patients with gastric cancer who show increased levels of D-dimer and CA724 are likely to face a worse prognosis after surgical intervention. Therefore, we suggest that clinical doctors should pay attention to the initial levels of D-dimer and CA724 in each hospitalized gastric cancer patient and conduct a comprehensive evaluation. Patients with high levels of D-dimer and CA724 usually indicate worse tumor blood and microenvironment status, and require more attention during treatment and follow-up. This insight may inform innovative strategies for postoperative adjuvant therapy in gastric cancer patients, including the potential use of anticoagulant treatment, aimed at optimizing survival and promoting advancements in precision oncology.” According to the latest results, the combination of D-dimer and CA724 can better evaluate the tumor microenvironment of patients, assess their prognosis, and provide new ideas and theoretical support for postoperative adjuvant therapy.

(All modifications have been highlighted prominently in the manuscript-mark)

Thank you for your professional and valuable suggestions on this manuscript! These suggestions have greatly improved the readability of this manuscript. If you have any additional questions or comments, please do not hesitate to contact us. We are very willing to make revisions and improvements based on your suggestions.

Reviewer2.

Thank you very much for your contribution to this manuscript!

Editor.

Thank you very much for your contribution to this manuscript! If you need our cooperation in the future, please feel free to contact us at any time. We are very willing to provide additional information!

---

## [Editor Report · Decision Letter 2]

16 Feb 2025

Evaluating the predictive significance of D-dimer in conjunction with  CA724 for the postoperative outcomes in gastric cancer: A retrospective cohort analysis.

PONE-D-24-54059R2

Dear Dr. Lu,

We’re pleased to inform you that your manuscript has been judged scientifically suitable for publication and will be formally accepted for publication once it meets all outstanding technical requirements.

Kind regards,

Xingliang Dai

Academic Editor

PLOS ONE

Additional Editor Comments (optional):

Acceptable manuscript
---

## [Editor Report · Acceptance letter]

PONE-D-24-54059R2

PLOS ONE

Dear Dr. lu,

I'm pleased to inform you that your manuscript has been deemed suitable for publication in PLOS ONE. Congratulations! Your manuscript is now being handed over to our production team.

Kind regards,

on behalf of

Dr. Xingliang Dai

Academic Editor

PLOS ONE